# Impacts of Gum Arabic and Polyvinylpyrrolidone (PVP) with Salicylic Acid on Peach Fruit (*Prunus persica*) Shelf Life

**DOI:** 10.3390/molecules27082595

**Published:** 2022-04-18

**Authors:** Mohamed A. Taher, A. A. Lo’ay, Mostafa Gouda, Safaa A. Limam, Mohamed F. M. Abdelkader, Samah O. Osman, Mohammad Fikry, Esmat F. Ali, Sayed. Y. Mohamed, Hoda A. Khalil, Diaa O. El-Ansary, Sherif F. El-Gioushy, Hesham S. Ghazzawy, Aly M. Ibrahim, Mahmoud F. Maklad, Mohamed A. Abdein, Dalia M. Hikal

**Affiliations:** 1Agricultural Chemistry Department, Faculty of Agriculture, Mansoura University, El-Mansoura 35336, Egypt; mohamedtaher@mans.edu.eg; 2Pomology Department, Faculty of Agriculture, Mansoura University, El-Mansoura 35336, Egypt; 3College of Biosystems Engineering and Food Science, Zhejiang University, Hangzhou 310058, China; 4Department of Nutrition and Food Science, National Research Centre, Dokki, Giza 12422, Egypt; 5Food Science and Technology Department, Faculty of Agriculture, Assiut University, Assiut 71526, Egypt; limamsafaa@gmail.com; 6Department of Plant Production, College of Food and Agriculture, King Saud University, Riyadh 12372, Saudi Arabia; mohabdelkader@ksu.edu.sa; 7Horticulture Research Institute, Agricultural Research Center, Giza 12619, Egypt; ayatosman012@gmail.com (S.O.O.); sayed_h_11@yahoo.com (S.Y.M.); dr.alyibrahim70@gmail.com (A.M.I.); 8Department of Agricultural and Biosystems Engineering, Faculty of Agriculture, Benha University, Moshtohor, Toukh 13736, Egypt; moh.eltahlawy@fagr.bu.edu.eg; 9Department of Biology, College of Science, Taif University, Taif 21944, Saudi Arabia; a.esmat@tu.edu.sa; 10Department of Pomology, Faculty of Agriculture (EL-Shatby), Alexandria University, Alexandria 21545, Egypt; hoda.khalil@alexu.edu.eg; 11Precision Agriculture Laboratory, Pomology Department, Faculty of Agriculture (El-Shatby), Alexandria University, Alexandria 21545, Egypt; diaa.elansary@alexu.edu.eg; 12Horticulture Department, Faculty of Agriculture (Moshtohor), Benha University, Moshtohor, Toukh 13736, Egypt; S.F.El-Gioushy@yahoo.com or; 13Date Palm Research Center of Excellence, King Faisal University, Hofuf 31982, Saudi Arabia; 14Central Laboratory for Date Palm Research and Development, Agriculture Research Center, Giza 12511, Egypt; 15Department of Horticulture, Faculty of Agriculture, Ain Shams University, Cairo 11566, Egypt; mahmoud_maklad@agr.asu.edu.eg; 16Biology Department, Faculty of Arts and Science, Northern Border University, Rafha 91911, Saudi Arabia; 17Nutrition and Food Science, Home Economics Department, Faculty of Specific Education, Mansura University, Mansoura 35516, Egypt; dr.daliahikal@mans.edu.eg

**Keywords:** peach, shelf-life, cell-wall-degrading enzymes, plant tissue, gum arabic

## Abstract

Peaches are grown in many Egyptian orchards for local and global fresh market sales. The interior fruit tissue breakdown (IFTB), often resulting in decayed peaches, is a severe problem during marketing. Therefore, to minimize FTB of peaches, in this study, gum arabic (GA) and polyvinylpyrrolidone (PVP) were mixed with different concentrations of salicylic acid (SA) (0, 1, and 2 mM) and were applied as edible coating to extend the shelf life of peach fruits. Mature peaches were selected and harvested when peaches reached total soluble solid content (SSC: 8.5%) and fruit firmness of about 47 N. Fruits were coated and stored at room temperature (26 ± 1 °C and air humidity 51 ± 1%) for 10 days during two seasons: 2020 and 2021. Fruit coated with GA/PVP-SA 2 mM showed a significant (*p* < 0.05) inhibition in degrading enzyme activities (CWDEs), such as lipoxygenase (LOX), cellulase (CEL), and pectinase (PT), compared to uncoated and coated fruits during the shelf-life period. Hence, cell wall compartments were maintained. Consequently, there was a reduction in browning symptoms in fruits by inhibiting polyphenol oxidase (PPO) and phenylalanine ammonia-lyase (PAL) activities. Thus, the fruit skin browning index showed almost no symptoms. The lipid peroxidation process and ionic permeability declined as well. The result suggests that, by applying GA/PVP-SA 2 mM as an edible coating, fruit tissue breakdown can be minimized, and the shelf life of peach can be extended up to 10 days without symptoms of tissue breakdown.

## 1. Introduction

The peach fruit (*Prunus persica* L. cv, ‘Early Sweet’), which belongs to the family Rosaceae, is one of the most popular fruits in the world and, in particular, in the Egyptian market due to its nutritive value and characteristic flavor [1]. Moreover, this fruit contains considerable amounts of bioactive pigments, such as anthocyanins, carotenoids, lutein, and β-cryptoxanthin [2]. Peach fruits lose water and wilt quickly as they progress through the ripening cycle. Therefore, the short postharvest life of peach fruit makes it unmarketable due to the extreme breakdown of interior tissues accompanied by microbial infections [3]. Interior tissue breakdown in peach fruit, indicated by flesh browning, is characterized by elevated neutral sugar and low amounts of cellulose and pectin, as well as diminished activity of pectin-hydrolyzing enzymes and cation binding, mostly calcium, in the cell wall [4]. Cold storage is proposed as the most appropriate technique to slow down the decay processes and preserve the fruit quality. However, peach fruits are sensitive to low storage temperatures, which limit their storage period [5].

Currently, some edible polymers are being extensively applied alone to prolong the shelf life and quality attributes of edible fruits [4,5]. In the case of peach fruit, the desirable effects of different *Aloe* gels on delaying ethylene production, ripening index, color development, and weight loss during storage of peach fruits at ambient temperature have been reported [6]. Additionally, *Aloe vera* gel alone has been documented as a suitable edible film for peach fruits regarding shelf life [7].

The blending of biodegradable polymers is one of the most valuable techniques to produce a new material with mechanically desirable properties in comparison with the individual polymers. In this respect, the prolonged shelf life of guava fruit by a cashew gum/carboxymethyl cellulose biopolymer blend has been reported [8]. The incorporation of blends of chitosan/PVA biopolymers and oxalic or ascorbic acids alone were, respectively, reported to alleviate skin browning in bananas [9] and to minimize the activities of cell-wall-degrading enzymes (CWDEs) of ‘superior seedling’ grapes during storage [10,11]. Furthermore, these blends inhibit the fungal infections [12] and preserve the quality of the fruit [13].

Gum arabic (GA) is one of the most common polysaccharides and is naturally extracted from the bark of *Acacia senegal*. The Food and Agriculture Organization of the United Nations (FAO) has permitted GA as a safe additive coating in food industries [14]. Moreover, a blended coating containing GA and chitosan was also stated to enhance fruit quality in banana [15,16]. Structurally, arabinogalactan is the main component of GA (80–90%) [17]. The characteristic properties of GA have explained its diverse applications, i.e., emulsification [18], stabilization [19], and micro-encapsulation [20]. GA is widely employed in the postharvest processing of edible fruits [21,22,23]. Polyvinylpyrrolidone (PVP) is a safe nonionic amorphous polymer with a high level of solubility in water [24]. PVP, with its useful properties, such as water solubility, absence of toxicity, film development, and adhesive power, is one of the best hopeful polymers for nanogels research [25]. PVP has been considered to be a prospective polymer with a great film-forming capacity for probable application in the production of coatings, cosmetics, detergents, plastics, medicine, and pharmaceuticals [26,27]. The United States Food and Drug Administration (FDA) has permitted PVP for various applications [28]. Acceptable daily intake of PVP has been documented as 0–50 mg/kg/day [29]. Food applications of PVP, including coating for fresh citrus fruits, clarification of beverages, binder for vitamin and mineral concentrates, binder for synthetic sweeteners, and dehydration of aqueous foods, such as orange and tomato juices, have been documented [30].

Salicylic acid (SA; also known as 2-hydroxy benzoic acid) is an endogenous growth regulator and a signal molecule that is critical for the induction of resistance to biotic and abiotic stress. In plants, it exists as a free phenolic acid and as conjugated forms that are constructed by hydroxylation, glycosylation, or methylation of the aromatic ring [31]. Earlier studies have stated the profits of preharvest and postharvest treatments with SA on different fruit quality features, such as ascorbic acid content, greater weight, and firmness in peach fruit [31] and grape fruit [32]; lower level of degradation of carotenoids in grape tissue color [33] and better chroma index; and higher total soluble solids, bioactive ingredients, and antioxidant activities and enhanced activity of some antioxidant enzymes in sweet cherry fruit [34,35]. Much research has stated the higher contents of total polyphenols and flavonoids in some SA-treated fruits, such as sweet cherries [34], peaches [36], and apples [37]. Moreover, numerous studies have stated the benefits of postharvest applications with SA in polymer coatings, such as chitosan/PVP in guava [31] and chitosan enriched by nanosized titanium dioxide particles in blackcurrant fruit [33].

Regarding public doubts about the undesirable effects of synthetic fungicides on human health and the environment, there is continuing research into new substitutes for the application of synthetic chemicals. One of the alternatives might be the use of SA, which has revealed antifungal properties on some fruits and other plants [38]. For instance, Babalar et al. [38] stated the effectiveness of SA against the decay caused by *Botrytis cinerea* in strawberry fruit. It is also recognized that SA has fungicidal properties on the brown rot of sweet cherry fruit, which is caused by *Monilinia fructicola* [39].

There are no available data in the literature concerning the use of a biopolymer blend of GA/PVP supplemented with salicylic (SA) acid as an edible coating. Thus, the present study aimed to evaluate the ability of GA/PVP/SA edible coating formulations in minimizing the incidences of tissue breakdown in peach fruits’ ‘Early Sweet‘ during the trading period.

## 2. Results

### 2.1. Fourier Transform Infrared Analysis (FT-IR)

Figure 1 shows FT-IR spectra of pure SA, GA, PVP, and the blends of GA/PVP and GA/PVP–SA 2 mM. The spectra showed characteristic bands of vibrations of the functional groups formed in the prepared blends. The IR spectrum of salicylic acid (Figure 1) reveal the absorption bands at 3238, 3063, 1659, 1612, 1577, 1483, 1443, and 1297 cm^−1^. Meanwhile, the peaks at 3422, 2927, 1654, 1429, 1030, and 776 cm^−1^ were noted for GA. The IR spectrum of PVP shows the characteristic peaks observed at 3450, 1656, 1435, 1289, and 845 cm^−1^. Moreover, the characteristic bands at 3451, 2926, 1294, and 776 cm^−1^ were found in the GA/PVP blend (Figure 1). The FT-IR spectrum of GA/PVP–SA 2 mM has prominent bands at 3528, 3240, 1660, 1612, 1483, 1296, and 760 cm^−1^ wavenumbers.

### 2.2. Evaluation of Fruit Properties: Water Loss %, Skin Browning Index (SBI), Hue Angle, and Firmness (N)

GA/PVP blends as edible coatings possessed significant effects on weight loss of peach fruits compared with control uncoated fruits (Table 1). The GA/PVP–SA 2 mM coating treatment recorded the lowest percentage of water loss (%) throughout the storage period. The skin browning index of control uncoated fruits recorded a slow raise until the sixth day, followed by a sharp increase and, therefore, reached the highest value (3.75) at the end of the storage period (Table 1). Meanwhile, its values did not alter throughout the experiment period in fruits treated with GA/PVP/SA composite blends. An acceptable low level of SBI was also obtained when peach fruit was treated with a GA/PVP blend.

GA/PVP blends had a significant impact on the hue angle (color appearance parameter) of peach fruits compared with control uncoated fruits (Table 1). Throughout the experiment, fruits treated with GA/PVP blends developed higher hue angle values in comparison with the non-coated fruits. The color appearance parameter (ho) of fruits ultimately drops; however, the level of decrease in the fruits coated with GA/PVP blends is much lower than the uncoated fruits. Moreover, GA/PVP–SA 2 mM treatment displayed the highest ho value (17.96) on the 10th day of the storage period (Table 1).

Results indicated that GA/PVP coating blends have a significant impact on peach fruit firmness compared with control uncoated fruits (Table 1). The fruit firmness gradually decreased all across the storage period; however, the grade of decrease in the fruits coated with GA/PVP blends is much lower than untreated fruits. The GA/PVP blend enriched with 2 mM salicylic acid in this study resulted in the lowest loss of firmness of peach fruits, which was noticeably more effectual throughout the short period (10 days) of storage at 25 °C (Table 1).

### 2.3. The Fruit Skin Browning Variables: Total Phenol (TPs), Flavonoids (FLs), and Browning Enzyme Activities

During 10 days of storage at 25 °C, the contents of TPs and TFs of uncoated fruits were significantly lower than those of GA/PVP treatments (Table 2). Noticeably, the phenolic contents regularly diminished in coated and uncoated fruits throughout the shelf-life period. Moreover, the GA/PVP–SA 2 mM treatment recorded the highest contents of TPs and TFs during the 10 days of storage time of peach fruits. The treatment of GA/PVP–SA 2 mM showed the highest TP and FL contents at the end of the experiment (TP, 79.89; FL, 23.41 mg 100 g^−1^), while the control fruits displayed a progressive decline in phenolic contents on the 10th day (TP, 55.90; FL, 12.38 mg 100 g^−1^ FW), respectively.

The results showed that the activities of PPO and PAL increased in both uncoated and coated peach fruits throughout the storage period (Table 2). Nevertheless, PPO and PAL activities in coated fruits increased slowly in comparison with control fruits. After 10 days of the experiment, minimum activities of PPO and PAL were noted in fruits dipped in GA/PVP–SA 2 mM (0.35 and 10.22 U min^−1^ mg protein^−1^), while the highest activities of PPO and PAL were recorded in uncoated fruits (0.65 and 24.31 U min^−1^ mg protein^−1^, Table 2, respectively).

### 2.4. The Activities of Cell-Wall-Degrading Enzymes (CWDEs)

Table 3 shows the alteration in the activities of CWDEs, i.e., LOX, CEL, and PG (U min^−1^ mg protein^−1^), throughout the shelf-life period (days) for ‘Early sweet’ peach fruit. A significant interaction at *p* ≤ 0.001 was observed when coating treatment of GA/PVP–SA and storage periods (days) were considered as a factor. Noticeably, all the activities of CWDEs increased gradually up to the sixth day of the experiment. Moreover, both LOX and PG enzymes continued to increase in activity until the 10th day of postharvest life; however, the CEL activity decreased. As presented in Table 3, the control treatment presents the highest significant activities of LOX (2.28 U min^−1^ mg protein^−^^1^), CEL (17.85 U min^−1^ mg protein^−1^), and PG (2.86 U min^−1^ mg protein^−1^) on the 10th day. Conversely, the GA/PVP–SA 2 mM treatment presents the highest significant reduction in CWDEs, i.e., LOX (0.99 U min^−1^ mg protein^−1^), CEL (9.64 U min^−1^ mg protein^−1^), and PG (0.43 U min^−1^ mg protein^−1^), at the end of the experiment.

### 2.5. Cell Membrane: Lipid Peroxidation (MDA; μM mg^−1^ FW) and Electrolyte Leakage (EL)

Experimental data of MDA and EL% as indicators of membrane disruption are presented in this study. Table 4 exhibits a significant interaction at *p* < 0.001 between postharvest time (days) and GA/PVP–SA treatments. Perceptibly, both cell membrane termination and MDA and EL% increased gradually over all treatments up to the 10th day of the postharvest period, contrasting with the initial rates at harvest time. Moreover, the control treatment showed more rapid MDA accumulation and ion permeability percentage during the postharvest period compared to other coating treatments. The coating treatment of GA/PVP–SA 2 mM is recognized as the most valuable one in minimizing the alternations in cell membrane parameters, where it recorded the values of 0.32 μM g^−1^ FW and 13.24% for MDA and EL%, respectively, on the 10th day of storage time.

### 2.6. Fruit Ethylene and Respiration

Figure 1 shows the differences in ethylene production and respiration rates of peaches during storage in shelf-life conditions. Each of the two gases rose to a maximum on the second day that is higher than the initial value. During the shelf-life continuance, ethylene increases clearly to a maximum of three times and, for the respiration, 2.5 times on the second day for the control treatment. The increases are independent according to the treatments. GA/PVP–SA 2 mM presented more inhibition in both ethylene and respiration throughout the experiment (Table 5). It recorded 11.17 and 14.57 mg kg^−1^ h^−1^ for both gasses on the second day of the shelf-life period. Consequently, it minimizes both ethylene and carbon dioxide production until up to the end of the storage time (Figure 2).

### 2.7. Data Modeling

Linear regression analysis was performed to predict the properties of the treated fruit with the best treatment (AG/PVP–SA 2 mM). It can be concluded from Table 6 that R^2^ values of WL%, firmness, MDA, EL%, and the enzyme activities of PAL, PPO, and PT are greater than 0.90, meaning that the linear model could be properly used for forecasting the characteristics of the treated peach with AG/PVP–SA 2 mM at different storage periods from 0 to 10 days. Meanwhile, the linear model cannot be useful for predicting ethylene, SBI, and CEL due to its lower R^2^ values, which were less than 0.7.

## 3. Discussion

The IR spectrum of salicylic acid (Figure 1) reveals the absorption bands ascribed to the stretching vibrations of O–H bonds of phenyl hydroxyls (3238 cm^−1^), and the stretching (a shoulder at 3063 cm^−1^) and bending (1297 cm^−1^) vibrations of C–H bonds of aromatic rings, C=O bonds (1659 cm^−1^), and C=C bonds of benzene rings (1612, 1577, 1483, and 1443 cm^−1^). Overall, the FT-IR spectrum of SA in this study agreed to a large extent with that obtained by [40].

For GA, the broad peak observed at 3422 cm^−1^ is ascribed to OH groups of the carbohydrate structure. Meanwhile, the major IR bands observed at 2927 cm^−1^ were assigned to the vibrational modes of C–H groups. Peaks present in the spectra at 1654 and 1429 cm^−1^ correspond to the occurrence of the carboxylic groups. It is well known that carboxylic acids display a characteristic OH in-plane bending band at 1430 cm^−1^ [41]. Therefore, the peak found at 1429 cm^−1^ wavenumber may be due to the symmetrical stretching of uronic acid carboxylates in the structure of GA. The peaks found between 800 cm^−1^ and 1200 cm^−1^ represented C–C, C–O, and C–O–C stretching and C-OH and C–H bending modes of the polymer backbone. Bands detected in the spectra at 776 cm^−1^ may be assigned to the 1–4 linkage of galactose and 1–6 linkage of mannose [41]. The peaks observed at 1030 and 879 cm^−1^ in the FT-IR curve of GA may be assigned to arabinogalactan. In the present work, the bands found between 700 and 500 cm^−1^ were attributed to the pyranose rings. Our results concerning the FT-IR spectrum of GA were in accordance with those obtained by [41].

For PVP, the peaks observed at 3450 cm^−1^ are assigned to hydroxyl stretching, and the peaks at 1435 and 845 cm^−1^ correspond to the CH_2_ scissoring vibrations and CH_2_ bending, respectively. The peaks at 1656 and 1289 cm^−1^ are assigned to C=O stretching and C–N stretching. Our results concerning the FT-IR spectrum of PVP were in accordance with those obtained in [42].

In the GA/PVP blend, the C–N bending vibration from the PVP pyrrolidone structure was shifted to be at 1294 cm^−1^, while the peak at 776 cm^−1^ in this blend confirmed the presence of 1–4 linkage of galactose and 1–6 linkage of mannose related to GA. The FT-IR spectrum of GA/PVP–SA 2 mM has prominent bands at 3240, 1660, 1612, and 1483 cm^−^^1^ wavenumbers due to the presence of SA in the polymer blend. Meanwhile, the peak found at 1296 cm^−1^ might be related to the pyrrolidone structure of PVP. The board peak of hydroxyl groups related to GA shifts to the wavenumber of 3528 cm^−1^. Moreover, the shifted peak at 760 cm^−1^ related to the 1–4 linkage of galactose and 1–6 linkage of mannose might support the presence of GA in GA/PVP–SA 2 mM treatment.

Coating polymers are generally made of fats, proteins, and polysaccharides that inhibit water loss [15,43] and probably remain less prone to microbial attack [44,45]. The reason for decreasing water loss by GA/PVP blends in this study is principally due to the properties of GA arabinogalactan (80–90%) in the retaining of water, which prevents the loss of fruits’ water during the storage period [15]. Similarly, the authors of [6] found that two varieties of *Aloe* gels as a coating source were able to diminish water loss of palm and peach fruits compared with uncoated fruits. Polysaccharide biodegradable coatings can decrease water loss of the fruit tissues by forming a physical barrier around the fruit surface [41]. Overall, the efficiency of a polysaccharide polymer as a biodegradable coating critically depends on its physical properties [42]. It is thought that high molecular weight polysaccharides, such as GA, have huge mechanical properties [42]. PVP is a synthetic polymer that forms a hydrogel that can preserve a huge amount of water.

In addition, using PVP increases coating performance to maintain peach fruit quality and decline the water evaporation. Thus, the use of pure PVP is restricted. To overcome this difficulty, PVP is fabricated with GA in this study to obtain a blend possessing acceptable properties, i.e., forming a physical barrier and preserving a high amount of water [46].

The presence of phenolic ingredients increases cellular immunity due to their biological properties, such as antioxidant capacity and protection against fungi and bacteria [43]. GA/PVP–SA 2 mM presents a more effective treatment that maintains the phenolic load throughout the shelf-life period of peach fruits. Our findings agreed to a large extent with those obtained by [44], who found that 1.5 mM SA-treated peach fruits kept the higher significant amounts of ascorbic acid, TFs, and TPs and antioxidant capacity when compared to untreated fruits.

Enzymatic browning is a common phenomenon that can usually be detected in different fruits, which unfavorably affects the nutritional value and other quality attributes. This happens when the phenolic ingredients are oxidized by PPO to their quinone derivatives and, additionally, are oxidized to form melanin pigment accountable for the browning reactions [45]. In the present study, GA/PVP–SA coatings can decrease the incidences of skin browning in peach fruit by decreasing PPO and PAL activities. The effectiveness of GA/PVP coatings in decreasing the action of browning enzymes could be due to their supplementation with salicylic acid. In this respect, the authors of [31] stated that SA shows an important role in the inhibition of the activities of browning-related enzymes in guava fruits.

The impact of SA on phenolic contents and the activities of browning enzymes depend on some critical factors, such as type of stress, SA concentration, availability, plant variety, postharvest conditions, and so on. In this respect, postharvest treatment of peach fruit with SA at 2 mM recorded lower PPO activity, accompanied by higher activities of antioxidant enzymes during shelf-life storage [46]. Moreover, another study indicated that salicylic acid alleviated chilling incidence, reduced PAL activity, and preserved phenolics and antioxidant capacity in pomegranate fruit in the postharvest period [47]. Showing different behavior, hot salicylic acid preserved higher anthocyanin and total phenolic contents in the arils of pomegranate during postharvest storage at 4 °C for nearly two weeks via diminishing PPO activity in combination with better PAL activity [7,48,49]. Similarly, improved total phenol content in SA- and calcium-chloride-treated cornelian cherry fruits may be attributed to higher PAL activity [6]. On the contrary, the application of SA in sponge gourds significantly decreased TPs due to its ability to inhibit the activities of PAL and PPO browning enzymes [50,51]. In our study, however, SA incorporated into the GA/PVP coating can decrease the activities of PPO and PAL browning enzymes. The increased amounts of polyphenols in GA/PVP–SA-treated fruits might be due to the higher PAL/PPO ratio. Overall, the higher accumulation of phenolic compounds in fruit tissues without undesirable browning incidences is largely correlated to a higher PAL/PPO ratio [48].

Skin browning is a common problem for extending the postharvest life of peach fruit, which is most sensitive to mechanical injury. Browning is chiefly caused by the enzymatic oxidation of endogenous phenols into quinones [47]. SBI weighs the clarity of the brown color and is reflected as a critical factor for examining the types of browning in fruits [52]. It was detected that SBI in the control peach fruits had extensive variations and specifically increased after 6 days of storage. This finding agreed with the obtained highest level of PPO activity in the control group throughout the storage period (Table 1). Meanwhile, the coated fruits did not display any significant alteration over the whole period of storage, reflecting the ability of GA/PVP blends to prevent browning incidences as a result of their ability to decrease the activity of browning enzymes.

The acceptable color and appearance of peach fruit is the critical factor for its friendly marketing. The hue angle of vegetables and fruits was mostly affected by the coating treatment. Moreover, coated peach fruits’ color should stay parallel to a fresh one by combating any deterioration of color during storage. In this regard, the little rate of decrease in hue angle of peach fruits coated with GA/PVP blends in comparison to uncoated fruits reflects the effectiveness of these polymer composites in avoiding color rapid deteriorations. Our results agreed with [7], who showed that peach fruits coated with *Aloe vera* gel under shelf-life storage over 30 days had a rate of diminished hue angle that was much slighter than the uncoated fruits. It could be suggested that GA/PVP blends can prevent color rapid deterioration [48]. The proper role of SA in polymer blends in inhibiting the enzymes that cause tissue softness may reflect the retardation of color changes in peach fruit treated with GA/PVP–SA 2 mM.

During the ripening process, fruit firmness decreased gradually due to the induction of the activities of CWDEs. In this study, GA/PVP formulations could preserve peach fruit firmness, especially GA/PVP–SA 2 mM. The literature data revealed a discrepancy in the impact of different edible coatings on peach fruit firmness, validating the proper role of SA in the present study. For instance, it was found that *Aloe* gels had no effect on the level of firmness in peach fruit stored at 20 °C for six days; its values gradually decreased in coated and uncoated fruits alike, with no significance [49]. In contrast, a significant impact beginning from the 10th day of *A. vera* coating film decreasing the firmness loss of peach fruits stored at 4 °C has been recently reported [7].

The present data elucidated the efficient role of GA/PVP biopolymer coatings in diminishing the activities of PG, CEL, and LOX (Table 3). Thus, the softness retardation of peach fruits treated by GA/PVP coatings in this study may be due to the property of GA and SA alone to inhibit the activities of CWDEs, which preserve firmness for as long as possible. In this respect, the inhibitory effect of GA/chitosan (10:1) composite against the activities of CWDEs, and thus keeping the level of firmness in stored banana fruit for as long as possible, has been documented [15]. Meanwhile, the ability of chitosan-based polymer blends to suppress the activities of CWDEs in guava fruits was significantly increased in the presence of SA, thereby reducing tissue breakdown and fruit water loss and preserving the level of firmness in three phases of fruit maturity [31].

The plant cell wall is a complex reticulate structure, which consists of structural proteins, pectin, cellulose, and hemicellulose [50]. Pectin is the chief element in the cell’s primary wall and the middle lamella and can tie cells together, similarly to ‘glue’. During ripening, fruit firmness decreases regularly due to the initiation of the activities of CWDEs, such as PG, LOX, and CEL. Many applications have been stated to reduce ethylene production and, thereby, prevent the activity of the enzymes, delaying softness.

The hydrolysis of pectin is catalyzed by related enzymes comprising pectin methylesterases, PG, pectate-lyase, and β-galactosidase, of which PG has been proposed to act as a vital role [51]. Moreover, gene expression of PGs during softening of two peach fruit cultivars with different softening features has been recently identified [52]. Thus, in this study, the activity of PG was examined as an appropriate indicator of cell wall pectin hydrolysis. The maximum obtained level of PG activity in control uncoated fruits in this study clarified their rapid ripening and softening.

Cellulases (E.C. 3.2.1.4) hydrolyze β-1,4 linkages of cellulose, cellobiose, and cellodextrin. Generally, they are multienzyme complexes having endo-1,4-β-glucanase, β-glucosidase, and cellobiohydrolase activity [53]. In fruits, cellulase activity is mostly correlated to softening physiological processes during maturation. The highest obtained level of CEL activity in control untreated fruits in this study elucidated their rapid ripening and loss of firmness. A high correlation between the huge level of cellulase activity and the minimum level of firmness in fruit tissues has been stated [54]. Enzymatic depolymerization of hemicellulose plays a key role in fruit maturation, leading to the disassembly of hemicellulose and the cellulose network and a reduction in fruit firmness [55]. Based on the obtained results, the increase in CEL activity in coated and uncoated fruits until the 6th day may be due to the disassembly of the hemicellulose/cellulose network. The decline in CEL activity after the 6th day in all treatments could be due to the extensive alternation in the hemicellulose structure.

In this study, the GA/PVP biopolymer coating efficiently diminished the activities of PG and CEL. These desired effects progressively increased in the presence of salicylic acid in the polymer composites. Inhibition of fruit CWDEs and preservation of firmness using edible coating formulations have been noted by several researchers. In this respect, Dave et al. [56] found that the formulations depend on hydroxyl-propyl methylcellulose; soy protein isolate and olive oil had an inhibitory effect on the activities of enzymes related to fruit softening, including pectin methylesterase, β-galactosidase, and PG, in pears stored at 28 °C. Srivastava and Dwivedi [44] found that 1 mM SA was able to delay softening by decreasing the activities of PG, CEL, and xylanase in bananas. Moreover, supplementation of chitosan-based polymer blends with SA has been reported to decline the activities of CWDEs in guava fruit [31]. Overall, earlier findings explained the role of SA application in the enhancement of the activity of polymer coatings.

Lipoxygenase (LOX, EC 1.13.11.12) is an enzyme that commonly exists in plant tissues, which activates the oxidation of polyunsaturated fatty acids to form corresponding hydroperoxides. The fatty acid hydroperoxides formed by the activity of LOX are possibly harmful to membrane function by initiating increased rigidity [47]. It also performs positively through its function in the development of defense-related signaling molecules [57]. Lipoxygenases possess some applications in food technology, such as aroma production and bread making; they also have undesirable effects, including off-flavor and color changes in different foods [47]. In this study, GA/PVP/SA biopolymer coatings effectively diminished the activity of LOX. This desired effect could be due to the presence of salicylic acid in the polymer composite. In this respect, the authors of [47] found a reduction in the expression of *DkLOX_3_* by SA, concomitant with the preservation of fruit firmness, inhibition of weight loss, and ethylene production during persimmon fruit storage. Moreover, the inhibitory effect of nitric oxide on LOX and ethylene biosynthesis in the shelf life of peach fruit has been previously reported [58]. The highest level of LOX activity of uncoated peach fruits at the end of the storage period in this study agreed with the obtained results regarding water loss and softness. Overall, the activity of LOX and other lipolytic enzymes increases during senescence [59], the earlier activity causing the leakage of membrane polyunsaturated fatty acids (PUFAs) that can act as a substrate for LOX. In response to wounding or senescence, LOX could be complicated undesirably through contribution to autocatalytic peroxidation reactions [60]. The resultant hydroperoxides can induce tissue injury through protein synthesis inactivation and dysfunction of cellular membranes. Lastly, the inhibitory effect of GA/PVP/SA biopolymer coatings on lipolytic enzymes in this study will offer new indications for exploring the roles of *LOX* in delaying peach fruit ripening and preserving firmness.

MDA is the product of lipid peroxidation, and its accumulation is indicative of cell membrane degradation. It was previously described that the increased amount of MDA is principally due to the increased activity of LOX [61]. Our results concerning lipid peroxidation and ion permeability suggest that GA/PVP–SA is a promising tool for avoiding postharvest oxidative damage.

The usage of biodegradable coatings alone or incorporated with bioactive additives in fruit postharvest technology significantly reduces the accumulation rate of MDA, which preserves the functions of cellular membranes and, thus, also reduces cell permeability rates [62]. This could be interpreted by the facilitation of coating material creating a barrier to the oxygen responsible for lipid peroxidation, hence maintaining membrane integrity [62]. Besides, GA was previously reported to delay ripening and consecutively preserve the antioxidant status of tomatoes up to three weeks after harvest [14]. Furthermore, the presence of SA in chitosan-based polymer blends may play a vital role in the inhibition of tissue breakdown by inhibiting the activities of CWDEs [31,57], as well as ethylene gas production and respiration [22]. Salicylic acid was also stated to reduce fruit senescence during shelf life [1,36].

The respiratory performance of GA/PVP/SA-coated peach fruits (Table 5) presented delayed the attainment of respiratory climacteric, and the respiratory magnitude was also found to be significantly (*p*  <  0.05) lower compared to uncoated fruits after the second day until the end of the storage period. Polymer coatings acted as a barrier film, providing a different internal atmosphere and a selective membrane for permeation of ethylene in and out of the fruit, as well as diminished production of ethylene by the fruit tissues [31]. Moreover, the inhibitory action of SA on the conversion of 1-aminocyclopropane-1-carboxylic acid (ACC) into ethylene via decreasing the activity of ACC oxidase has been reported [63].

The influence of SA on ethylene production was studied on different fruits, i.e., tomato [64], apple [65], and Selva strawberry fruit [63]. Lastly, a coating treatment of GA/PVP–SA 2 mM effectively decreased ethylene production in peach fruit, which might be explained by the role of SA in minimizing the respiration rate by increasing the energy charge [38].

## 4. Materials and Methods

### 4.1. Fruit Material

The present study was conducted throughout the season (2020 and 2021) on peach fruit (*Prunus persica*) ‘Early Sweet’ cv. Trees were planted in a commercial orchard in Meet-Gamer province, Egypt, in clay soil that is 11 years old. All trees were grafted on ‘Nemaguard’ rootstock at 4 × 4 m^2^ plantation distance. The trees were also pruned in an open vase shape under drip irrigation. The ranch administration program was connected by the proposals of the Egyptian Agricultural Ministry. Fruits were selected from the shaded side of trees in the same uniform at full maturation in May [66]. At harvest time, the fruit was selected when fruits reached SSC 8.5% and fruit firmness 47 N. A total of 360 fruits were harvested and divided into two main lots. The first lot contained 180 fruits dedicated to non-distractive measurements. Every treatment contained 45 fruits (15 fruits per replicate).

### 4.2. Gum Arabic and Coating Protocol

Gum arabic (GA) powder was supplied by El-Gomhoria Company, Cairo city, Egypt. It was prepared by adding 60 g GA to 500 mL distilled water. Thereafter, the solution was stirred by hot magnetic stirring at 55 °C for 60 min. Then, the solution was cooled at room temperature and filtered to remove impurities using a fine piece of cloth [14]. To prepare PVP (K-20 polymer, Ashland company, Shanghai, China), 15 g was dissolved in 500 mL of distilled water under magnetic stirring until dissolved. The polyvinylpyrrolidone (PVP) was supplemented with the GA solution in a ratio of (1:1) to form a coating mixture with desirable mechanical and physical properties. The resultant GA/PVP blend was then stirred for one hour. The final concentrations of GA and PVP in the resultant blend were 6% and 1.5%, respectively. To increase the covering mixture effectiveness, SA was supplemented to the mixture at three concentrations (0, 1, and 2 mM). Finally, the blended mixture solution was prepared in three main stocks GA/PVP–SA (0 mM), GA/PVP–SA (1 mM), and GA/PVP–SA (2 mM), each in a volume of 2 L. Afterward, fruits were soaked in the prepared mixtures and stored at 24 ± 3 °C and air humidity at 51 ± 1%.

### 4.3. Fourier Transform Infrared Analysis (FT-IR)

For identification of the functional groups and chemical bonding in pure SA, GA, PVP, and the dry powders of blend solutions of GA/PVP and GA/PVP-SA 2 mM, 1 mg of each sample was mixed with 300 mg of fine potassium bromide (KBr). The thin pellets were prepared by pressing with the hydraulic pellet press and were then subjected to Fourier transform infrared spectrophotometer (FTIR, Jinan City, China ) in the range of 500–4000 cm^−1^ at a resolution of 4 cm^−1^ [40,41,42].

### 4.4. Water Loss %, Skin Browning Index, and Fruit Color Hue Angle

Peach fruit samples (15 fruit per replicate) were weighed at harvest time up to the end of the experiment time interval (2 days). Weight loss % was computed based on the initial value at harvest time [31].

As for fruit skin, the browning parameter was determined by browning spots visually during shelf-life duration. The browning scale was identified in five categories [67]. The categories are: 1 = no brown spots; 2 = slight browning; 3 = moderate browning spot; 4 = severe browning symptoms; and 5 = very severe symptoms. Meanwhile, the fruit color hue angle parameter was assessed according to the RGB protocol [9].

Fruit firmness was measured as the necessary force to penetrate the tissue using the fruit texture Effegi penetrometer (Effegi, 48,011 Alfonsine, Alfonsine, Italy) [33].

### 4.5. The Browning Parameters: Total Phenols (TPs) and Total Flavonoids (TFs)

TPs content of peach fruit samples was evaluated using Folin–Ciocalteu reagent and gallic acid as a standard phenolic compound. TPs content was spectrophotometrically estimated at 750 nm. Data were represented as mg 100 g^−1^ fresh weight (FW) as gallic acid equivalents (GAE) [34]. Meanwhile, the flavonoid content was recorded in time intervals throughout the experimental time, and the data are shown as mg 100 g^−1^ FW catechin equivalent (CE) [35].

### 4.6. Extraction of Browning Enzyme

A gram of fruit sample in time was homogenized with 5 mL of Tris-HCl buffer (pH 7.0; 20 mM). Then, the mixture was centrifuged at 4 °C (15,000 rpm, 5 min) and the resultant supernatant was kept at −20 °C until further steps.

Polyphenol oxidase (PPO) (*EC: 1.14.18.1*) was assessed according to the procedure of [37] by mixing 0.5 mL catechol (500 mM) and 2 mL 0.05 M phosphate buffer, pH = 7.0, to 100 µL supernatant and incubated for 2 min at 24 °C. The enzyme activity was spectrophotometrically noted at 398 nm within 3 min. The enzyme activity was stated as units U min^−1^ mg protein^−1^.

Phenylalanine ammonia-lyase (PAL) (*EC: 4.3.1.24*) activity was assessed spectrophotometrically at 290 nm according to the steps described by [37] after its extraction from one gram of the fruit sample with 4 mL of 200 mmol L^−1^ boric acid buffer (pH 8.8). The activity of PAL was lastly expressed as units U min^−1^ mg protein^−1^.

### 4.7. The Activities of Cell-Wall-Degradation Enzymes (CWDEs) and Fruit Firmness

For enzyme extraction and assay, three grams of peach tissue was homogenized in 10 mL Tris-HCL (0.02 M, pH = 7.0) containing EDTA (20 mM), cysteine-HCL (0.02 M), and Triton X-100 (0.05%). Then, centrifugation was carried out at 15,000× *g* for half an hour at 4 °C. Lastly, the supernatant was kept at −20 °C until the enzymatic assessments [68]. The supernatant protein concentration was estimated according to Bradford’s method [69].

The assay of polygalacturonase (PG) activity was evaluated according to the scheme designated by [39]. Volumes of 400 μL of Na-acetate buffer (0.2 M, pH 4.5), 600 μL polygalacturonic acid (PGA, 1%, pH 4.5), and 200 μL sodium chloride (0.2 M) were successively added to 100 μL of the supernatant, and the resultant mixture was maintained at 37 °C for 60 min. Then, 6 mL 3,5-dinitro salicylic acid (DNS) was added. The reaction was blocked by heating at 85 °C for 15 min. Finally, 1 mL of 40% sodium potassium tartrate was added. In control samples, the substrate was added after heating. The formed reducing groups were assessed against *D*-galacturonic acid after assessing the absorbance at 540 nm. PG activity was expressed as U min^−1^ mg protein^−1^.

Assay of cellulase (CEL): the activity of CEL was evaluated according to the method described by [43]. The clear supernatant (250 μL) was added to the reaction mixture (500 μL carboxymethyl cellulose (1.5%) in 250 μL sodium acetate buffer (0.1 M, pH = 5.0)). The reaction mixture was incubated at 37 °C for 12 h. Then, 3 mL DNS was added, and the reaction was blocked by heating at 85 °C for 10 min. Finally, 1 mL of 40% SPT was added. In the control tube, the substrate was added after heating. The resultant reducing moieties were assessed against D-glucose after assessing the absorbance at 540 nm. CEL activity was expressed as U min^−1^ mg protein^−1^.

Lipoxygenase (EC: 1.13.11; LOX) activity was evaluated relating to the scheme [43]. Lipoxygenase was extracted after homogenizing 2 g of peach fruit with 6 mL of extracting buffer (0.6 mL potassium phosphate buffer (0.5 M, pH = 7.8), 0.6 mL sodium-EDTA (0.01 M, pH = 7), and 4.8 mL of PVPP (2% *w*/*w*)). The mixture was then centrifuged for 20 min at 17,000 rpm at 4 °C. The resultant supernatant (0.1 mL) was mixed with 2.8 mL of sodium phosphate buffer (0.1 M, pH = 6), and then 0.1 mL of linoleic acid sodium salt (0.005 M) was added. The activity of LOX was observed according to the resultant rise in absorbance at 234 nm due to hydroperoxide formation. The activity of LOX was expressed as U min^−1^ mg protein^−1^.

Two equidistant readings were taken in the equatorial region of each fruit. The results were expressed in Newtons (N) [31], and total soluble solid content (SSC%) was measured utilizing Portable Digital Refractometer (RFT-PD Series, Jinan City, China) [70].

### 4.8. Lipid Peroxidation and Cell Membrane Permeability

Malondialdehyde (MDA) was determined according to the method in [71]. One gram of peach pulp tissue was homogenized with 10 mL of 5% (*w*/*v*) meta-phosphoric acid and 0.2 mL of 2% (*w*/*v*) ethanolic butylated hydroxytoluene (BHT). The resultant mixture was centrifuged at 15,000× *g* rpm for 15 min. Then, 1.0 mL of the resultant supernatant was added to 0.1 mL of 2% (*w*/*v*) BHT, 0.5 mL of 1% (*w*/*v*) of thiobarbituric acid (TBA) in 0.05 M NaOH, and 0.5 mL of 20% (*v*/*v*) hydrochloric acid. The tube mixture was incubated at 95 °C for 30 min. After cooling, the resultant chromogen was extracted by adding 0.8 mL of n-butanol, centrifugation was performed to separate the organic phase, and the absorbance of thiobarbituric acid reactive substances (TBARS) was read at 532 nm. The compound of 1,1,3,3-tetra-ethoxy-propane (Sigma) was used to prepare the calibration curves, and finally, micromoles of MDA per gram FW for the examined samples were estimated. The electrolyte leakage percentage rate (EL%) was evaluated by a conductivity ion meter. Data were exhibited as a percentage [67].

### 4.9. Ethylene and Respiration

Ethylene and respiration of peach were estimated on a fruit sample of 6 peaches after a 2-day interlude. Fruits were incubated and saved in 1000 mL glass jars sealed for a one-hour interval of the shelf-life term in time for all further analyses. Gas samples only from each of the jars around the fruit were extracted, and both ethylene and carbon dioxide were examined by gas chromatography techniques (GC) (Darmstadt, Germany). However, ethylene was monitored by GC-6000 Vega Series (Carlo Erba Ins., Milano, Italy) and carbon dioxide was measured by GC PBI-Dansensor Checkmate-9900 (Copenhagen, Denmark).

### 4.10. Statistical Analysis and Data Modeling

The parameters of water loss %, fruit skin color hue angle, and skin spot browning index were analyzed using one-way analysis using a complete block design when GA/PVP–SA treatments were considered as a factor. However, the distractive parameters were analyzed using two-way analysis using a factorial experiment in a completely randomized block design for chemical parameters when storage period (days) and GA/PVP–SA treatments were investigated as factors. The means of all examined treatments were compared using Duncan’s multiple range test at *p* < 0.05 using Co-Stat software package Ver. 6.303 (798 lighthouse Ave PMB320, Monterey, CA, USA). To forecast the characteristics of the treated fruits during the shelf life, a linear regression technique was performed, and the criterion of R^2^ > 0.9 was considered as a proper judge for the fitness of the model [72].

## 5. Conclusions

The outcomes obtained from this research show that the peach fruit coated with GA/PVP mixed with SA at a concentration of 2 mM gives a significant effect on diminishing fruit tissue breakdown during postharvest life. This result can be summarized into three points: First, the treatment inhibits the activities of CWDEs. Second, it decreases the lipid peroxidation process and ionic penetrability rate. Third, the coating treatment decreases the activities of browning enzymes, and therefore, the brown incidence color on the fruits decreases during the shelf-life period. It can be said that GA/PVP–SA 2 mM treatment can retard the deterioration of peach fruit during shelf life. Future studies for the development of information on the impact of GA/PVP–SA treatment to increase fruit storability under shelf-life conditions are needed.

## Figures and Tables

**Figure 1 molecules-27-02595-f001:**
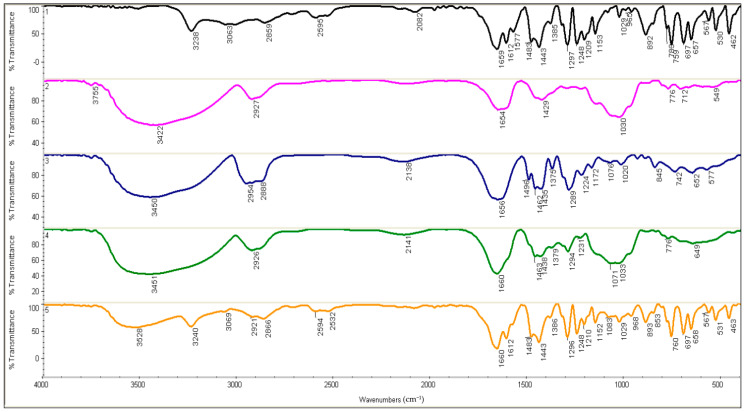
FTIR spectra of (1) SA, (2) GA, (3) PVP, (4) GA/PVP, and (5) GA/PVP/SA 2 mM. FT-IR, Fourier transform infrared spectroscopy; SA, salicylic acid; GA, gum arabic; PVP, polyvinylpyrrolidone.

**Figure 2 molecules-27-02595-f002:**
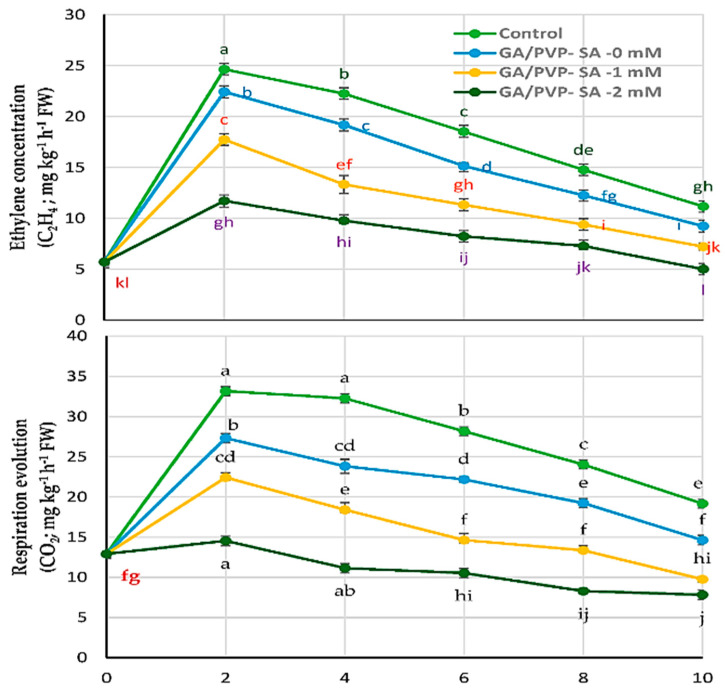
Effect of different treatments of gum arabic (GA) and polyvinylpyrrolidone (PVP) blended with salicylic acid (SA) at different concentrations as a mixture biopolymer coating treatment (GA/PVP–SA) on ethylene and respiration production rate of peach fruit ‘Early Sweet’ during shelf life at ambient air (29 ± 1 °C and air humidity 51%) during two seasons (2020 and 2021).

**Table 1 molecules-27-02595-t001:** Effect of different treatments of gum arabic (GA) and polyvinylpyrrolidone (PVP) blending with salicylic acid (SA) at different concentrations as a mixture biopolymer coating treatment (GA/PVP–SA) on physical properties of peach fruit ‘Early Sweet’ during shelf life at ambient air (29 ± 1 °C and air humidity 51%) during two seasons (2020 and 2021).

Treatments		Shelf-Life Period (Days)					
		D _0_	D _2_	D _4_	D _6_	D _8_	D _10_
**Water loss %**							
	Control	0.00 ± 0.000	3.14 ± 0.017 ^a^	11.97 ± 1.097 ^a^	18.47 ± 0.784 ^a^	27.35 ± 0.784 ^a^	31.30 ± 0.708 ^a^
	GA/PVP–SA 0 mM	0.00 ± 0.000	3.06 ± 0.012 ^b^	9.46 ± 0.309 ^b^	12.46 ± 0.652 ^b^	19.85 ± 0.311 ^b^	22.41 ± 0.584 ^b^
	GA/PVP–SA 1 mM	0.00 ± 0.000	3.02 ± 0.014 ^c^	7.86 ± 0.236 ^c^	9.67 ± 0.287 ^c^	16.71 ± 0.898 ^c^	18.28 ± 0.545 ^c^
	GA/PVP–SA 2 mM	0.00 ± 0.000	2.98 ± 0.008 ^d^	6.66 ± 0.193 ^c^	8.68 ± 0.193 ^d^	12.42 ± 0.506 ^d^	14.44 ± 0.578 ^d^
**Skin browning index**							
	Control	1.00 ± 0.000	1.02 ± 0.006 ^a^	1.16 ± 0.005 ^a^	1.60 ± 0.031 ^a^	1.99 ± 0.003 ^a^	3.75 ± 0.122 ^a^
	GA/PVP–SA 0 mM	1.00 ± 0.000	1.00 ± 0.000 ^b^	1.02 ± 0.014 ^b^	1.10 ± 0.008 ^b^	1.19 ± 0.008 ^b^	1.93 ± 0.061 ^b^
	GA/PVP–SA 1 mM	1.00 ± 0.000	1.00 ± 0.000 ^b^	1.00 ± 0.000 ^c^	1.00 ± 0.000 ^c^	1.01 ± 0.005 ^c^	1.08 ± 0.015 ^c^
	GA/PVP–SA 2 mM	1.00 ± 0.000	1.00 ± 0.000 ^b^	1.00 ± 0.000 ^c^	1.00 ± 0.000 ^c^	1.00 ± 0.000 ^c^	1.02 ± 0.005 ^c^
**Fruit color (hue angle)**							
	Control	5.81 ± 0.316 ^a^	8.67 ± 0.299 ^d^	12.28 ± 0.191 ^a^	12.80 ± 0.507 ^d^	9.59 ± 0.485 ^d^	9.29 ± 0.024 ^d^
	GA/PVP–SA 0 mM	5.81 ± 0.316 ^a^	9.85 ± 0.141 ^c^	13.09 ± 0.054 ^c^	14.19 ± 0.193 ^c^	13.46 ± 0.277 ^c^	12.57 ± 0.2137 ^c^
	GA/PVP–SA 1 mM	5.81 ± 0.316 ^a^	10.38 ± 0.175 ^b^	13.91 ± 0.200 ^b^	15.21 ± 0.153 ^b^	16.07 ± 0.256 ^b^	15.26 ± 0.021 ^b^
	GA/PVP–SA 2 mM	5.81 ± 0.316 ^a^	11.43 ± 0.158 ^a^	15.61 ± 0.300 ^a^	16.41 ± 0.515 ^a^	16.88 ± 0.106 ^a^	17.96 ± 0.229 ^a^
**Fruit firmness (*N*)**							
	Control	47.68 ± 0.655 ^a^	39.35 ± 0.498 ^ef^	32.38 ± 1.387 ^i^	29.50 ± 0.329 ^j^	25.71 ± 0.472 ^k^	19.30 ± 0.572 ^l^
	GA/PVP–SA 0 mM	47.68 ± 0.655 ^a^	40.63 ± 0.304 ^de^	38.03 ± 0.342 ^f^	35.18 ± 0.536 ^h^	32.19 ± 0.562 ^i^	29.59 ± 0.367 ^j^
	GA/PVP–SA 1 mM	47.68 ± 0.655 ^a^	44.25 ± 0.351 ^b^	42.81 ± 0.682 ^c^	39.43 ± 0.625 ^e^	36.55 ± 0.664 ^g^	31.62 ± 0.888 ^i^
	GA/PVP–SA 2 mM	47.68 ± 0.655 ^a^	46.70 ± 0.894 ^a^	44.41 ± 0.385 ^b^	42.59 ± 0.628 ^c^	40.80 ± 0.572 ^d^	39.47 ± 0.398 ^de^

Means in the same column that have different letter(s) are significantly different using Duncan’s multiple range test at *p* < 0.05 and ±SE (n = 3) replicates. Skin browning spot index was used to determine skin browning (1 = healthy; 2 = slightly brown; 3 = moderately brown; 4 = severe brown; and 5 = fully brown).

**Table 2 molecules-27-02595-t002:** Effect of different treatments of gum arabic (GA) and polyvinylpyrrolidone (PVP) blending with salicylic acid (SA) at different concentrations as a mixture biopolymer coating treatment (GA/PVP–SA) on skin browning elements of peach fruit ‘Early Sweet’ during shelf life at ambient air (29 ± 1 °C and air humidity 51%) during two seasons (2020 and 2021).

Treatments		Shelf-Life Period (Days)					
		D _0_	D _2_	D _4_	D _6_	D _8_	D _10_
**Total phenol content (TP; mg 100 g^−1^ FW)**							
	Control	86.58 ± 0.858 ^ab^	79.84 ± 0.896 ^de^	77.67 ± 0.360 ^fg^	66.80 ± 0.378 ^j^	60.86 ± 0.473 ^k^	55.90 ± 1.815 ^l^
	GA/PVP–SA 0 mM	86.58 ± 0.858 ^ab^	83.55 ± 0.619 ^c^	79.07 ± 0.330 ^ef^	75.39 ± 0.641 ^h^	70.99 ± 0.291 ^i^	67.56 ± 0.636 ^j^
	GA/PVP–SA 1 mM	86.58 ± 0.858 ^ab^	85.57 ± 0.569 ^b^	81.57 ± 0.715 ^d^	79.56 ± 0.402 ^ef^	77.63 ± 0.498 ^fg^	76.74 ± 0.531 ^gh^
	GA/PVP–SA 2 mM	86.58 ± 0.858 ^ab^	87.68 ± 0.200 ^a^	86.43 ± 0.646 ^ab^	81.66 ± 0.536 ^cd^	80.12 ± 0.360 ^de^	79.89 ± 0.564 ^de^
**Flavonoids content (FL; mg 100 g^−1^ FW)**							
	Control	33.69 ± 1.035 ^a^	29.47 ± 0.691 ^d^	26.44 ± 0.543 ^e^	17.34 ± 0.906 ^j^	15.40 ± 0.601 ^k^	12.38 ± 0.590 ^l^
	GA/PVP–SA 0 mM	33.69 ± 1.035 ^a^	30.91 ± 0.131 ^b^	29.38 ± 0.306 ^d^	21.41 ± 0.694 ^h^	19.61 ± 0.528 ^i^	16.92 ± 0.344 ^j^
	GA/PVP–SA 1 mM	33.69 ± 1.035 ^a^	31.53 ± 0.117 ^b^	30.57 ± 0.129 ^bc^	26.44 ± 0.543 ^e^	24.85 ± 0.626 ^f^	21.83 ± 0.294 ^h^
	GA/PVP–SA 2 mM	33.69 ± 1.035 ^a^	33.03 ± 0.389 ^a^	31.36 ± 0.317 ^b^	29.59 ± 0.442 ^cd^	26.40 ± 0.528 ^e^	23.41 ± 0.463 ^g^
**Phenylalanine ammonia-lyase activity (PAL; EC: 4.3.1.24 U min^−1^ mg^−1^ protein^−1^)**							
	Control	7.47 ± 0.269 ^j^	8.95 ± 0.076 ^h^	12.35 ± 0.513 ^ef^	15.71 ± 0.473 ^d^	16.95 ± 0.404 ^c^	24.31 ± 0.629 ^a^
	GA/PVP–SA 0 mM	7.47 ± 0.269 ^j^	8.55 ± 0.026 ^hi^	10.49 ± 0.348 ^g^	12.86 ± 0.566 ^e^	15.31 ± 0.294 ^d^	18.50 ± 0.440 ^b^
	GA/PVP–SA 1 mM	7.47 ± 0.269 ^j^	8.03 ± 0.030 ^ij^	8.92 ± 0.092 ^h^	10.58 ± 0.461 ^g^	11.86 ± 0.263 ^f^	12.68 ± 0.556 ^e^
	GA/PVP–SA 2 mM	7.47 ± 0.269 ^j^	7.85 ± 0.097 ^ij^	8.29 ± 0.023 ^hi^	9.03 ± 0.040 ^h^	9.85 ± 0.156 ^g^	10.22 ± 0.190 ^g^
**Polyphenol oxidase activity (PPO; EC: 1.14.18.1 U min^−1^ mg^−1^ protein^−1^)**							
	Control	0.26 ± 0.005 ^n^	0.34 ± 0.008 ^j^	0.44 ± 0.008 ^f^	0.52 ± 0.012 ^d^	0.58 ± 0.008 ^b^	0.65 ± 0.011 ^a^
	GA/PVP–SA 0 mM	0.26 ± 0.005 ^n^	0.31 ± 0.005 ^l^	0.39 ± 0.005 ^h^	0.46 ± 0.005 ^e^	0.54 ± 0.005 ^c^	0.59 ± 0.005 ^b^
	GA/PVP–SA 1 mM	0.26 ± 0.005 ^n^	0.29 ± 0.005 ^m^	0.33 ± 0.005 ^k^	0.37 ± 0.005 ^i^	0.40 ± 0.005 ^h^	0.43 ± 0.005 ^g^
	GA/PVP–SA 2 mM	0.26 ± 0.005 ^n^	0.26 ± 0.003 ^n^	0.29 ± 0.005 ^m^	0.31 ± 0.005 ^l^	0.33 ± 0.005 ^k^	0.35 ± 0.005 ^j^

The interaction between GA/PVP–SA coating treatments and shelf-life duration in days when both are considered as factors. Means in the same column that have different letter(s) are significantly different using Duncan’s multiple range test at *p* < 0.05 and ±SE (n = 3) replicates.

**Table 3 molecules-27-02595-t003:** Effect of different treatments of gum arabic (GA) and polyvinylpyrrolidone (PVP) blended with salicylic acid (SA) at different concentrations as a mixture biopolymer coating treatment (GA/PVP–SA) on cell-wall-degrading enzyme activities of peach fruit ‘Early Sweet’ during shelf life at ambient air (29 ± 1 °C and air humidity 51%) during two seasons (2020 and 2021).

Treatments		Shelf-Life Period (Days)					
		D _0_	D _2_	D _4_	D _6_	D _8_	D _10_
**Lipoxygenase activity (LOX; EC:1.13.11, U min^−1^ mg^−1^ protein^−1^)**							
	Control	0.84 ± 0.008 ^m^	0.93 ± 0.023 ^ij^	0.96 ± 0.021 ^hi^	1.31 ± 0.020 ^e^	1.94 ± 0.032 ^b^	2.28 ± 0.015 ^a^
	GA/PVP–SA 0 mM	0.84 ± 0.008 ^m^	0.88 ± 0.003 ^l^	0.92 ± 0.005 ^jk^	1.00 ± 0.012 ^g^	1.73 ± 0.043 ^c^	1.94 ± 0.023 ^b^
	GA/PVP–SA 1 mM	0.84 ± 0.008 ^m^	0.86 ± 0.008 ^lm^	0.88 ± 0.006 ^kl^	0.94 ± 0.005 ^ij^	1.11 ± 0.017 ^f^	1.35 ± 0.017 ^d^
	GA/PVP–SA 2 mM	0.84 ± 0.008 ^m^	0.84 ± 0.005 ^m^	0.86 ± 0.003 ^lm^	0.89 ± 0.005 ^kl^	0.93 ± 0.008 ^ij^	0.99 ± 0.005 ^gh^
**Cellulase activity (CEL; EC: 3.2.1.4 U min^−1^ mg^−1^ protein^−1^)**							
	Control	8.00 ± 0.063 ^i^	13.03 ± 0.777 ^f^	15.42 ± 0.666 ^e^	22.67 ± 1.353 ^a^	19.65 ± 0.619 ^b^	17.85 ± 0.543 ^c^
	GA/PVP–SA 0 mM	8.00 ± 0.063 ^i^	10.65 ± 0.450 ^g^	12.66 ± 0.538 ^f^	16.67 ± 0.390 ^d^	14.76 ± 0.459 ^e^	12.72 ± 0.592 ^f^
	GA/PVP–SA 1 mM	8.00 ± 0.063 ^i^	9.61 ± 0.363 ^h^	10.76 ± 0.453 ^g^	14.71 ± 0.509 ^e^	12.43 ± 0.738 ^f^	10.96 ± 0.383 ^g^
	GA/PVP–SA 2 mM	8.00 ± 0.063 ^i^	8.50 ± 0.100 ^i^	9.52 ± 0.272 ^h^	11.00 ± 0.455 ^g^	10.41 ± 0.057 ^gh^	9.64 ± 0.048 ^h^
**Pectinase activity (PT; EC: 3.2.1.15, U min^−1^ mg^−1^ protein^−1^)**							
	Control	0.27 ± 0.005 ^o^	0.36 ± 0.005 ^klm^	1.46 ± 0.014 ^d^	1.57 ± 0.005 ^c^	1.81 ± 0.033 ^b^	2.86 ± 0.067 ^a^
	GA/PVP–SA 0 mM	0.27 ± 0.005 ^o^	0.33 ± 0.005 ^mn^	0.40 ± 0.003 ^jk^	0.50 ± 0.005 ^i^	1.03 ± 0.040 ^f^	1.22 ± 0.063 ^e^
	GA/PVP–SA 1 mM	0.27 ± 0.005 ^o^	0.30 ± 0.003 ^no^	0.38 ± 0.003 ^jkl^	0.44 ± 0.005 ^j^	0.64 ± 0.008 ^h^	0.72 ± 0.005 ^g^
	GA/PVP–SA 2 mM	0.27 ± 0.005 ^o^	0.28 ± 0.003 ^no^	0.34 ± 0.005 ^lmn^	0.39 ± 0.006 ^jk^	0.42 ± 0.005 ^j^	0.43 ± 0.005 ^j^

The interaction between GA/PVP–SA coating treatments and shelf-life duration in days when both are considered factors. Means in the same column that have different letter(s) are significantly different using Duncan’s multiple range test at *p* < 0.05 and ± SE (n = 3) replicates.

**Table 4 molecules-27-02595-t004:** Effect of different treatments of gum arabic (GA) and polyvinylpyrrolidone (PVP) blended with salicylic acid (SA) at different concentrations as a mixture biopolymer coating treatment (GA/PVP–SA) on lipid peroxidation (MDA) and cell membrane leakage percentage of peach fruit ‘Early Sweet’ during shelf life at ambient air (29 ± 1 °C and air humidity 51%) during two seasons (2020 and 2021).

Treatments		Shelf-Life Period (Days)					
		D _0_	D _2_	D _4_	D _6_	D _8_	D _10_
**Lipid peroxidation (MDA; µM g^−1^ FW)**							
	Control	0.19 ± 0.005 ^n^	0.29 ± 0.012 ^j^	0.45 ± 0.008 ^e^	0.56 ± 0.020 ^c^	0.59 ± 0.011 ^b^	0.68 ± 0.014 ^a^
	GA/PVP–SA 0 mM	0.19 ± 0.005 ^n^	0.26 ± 0.005 ^k^	0.46 ± 0.005 ^e^	0.40 ± 0.005 ^f^	0.46 ± 0.005 ^e^	0.50 ± 0.005 ^d^
	GA/PVP–SA 1 mM	0.19 ± 0.005 ^n^	0.24 ± 0.005 ^l^	0.29 ± 0.005 ^j^	0.31 ± 0.012 ^ij^	0.37 ± 0.005 ^g^	0.39 ± 0.005 ^f^
	GA/PVP–SA 2 mM	0.19 ± 0.005 ^n^	0.21 ± 0.005 ^m^	0.22 ± 0.005 ^m^	0.25 ± 0.005 ^kl^	0.30 ± 0.005 ^j^	0.32 ± 0.005 ^i^
**Electrolyte leakage (EL%)**							
	Control	6.67 ± 0.273 ^p^	12.45 ± 0.631 ^kl^	18.75± 0.548 ^g^	24.89 ± 1.132 ^e^	35.20 ± 0.273 ^b^	52.88 ± 1.475 ^a^
	GA/PVP–SA 0 mM	6.67 ± 0.273 ^p^	11.09± 0.147 ^mn^	15.91 ± 0.419 ^i^	18.79 ± 0.275 ^g^	26.28 ± 0.273 ^d^	30.44 ± 0.419 ^c^
	GA/PVP–SA 1 mM	6.67 ± 0.273 ^p^	10.08 ± 0.037 ^n^	14.34 ± 0.328 ^j^	17.13 ± 0.273 ^h^	20.63 ± 0.273 ^f^	26.76 ± 0.539 ^d^
	GA/PVP–SA 2 mM	6.67 ± 0.273 ^p^	7.59 ± 0.117 ^p^	8.86 ± 0.150 ^o^	10.45± 0.273 ^mn^	11.62± 0.273 ^lm^	13.24± 0.541 ^jk^

The interaction between GA/PVP–SA coating treatments and shelf-life duration in days when both are considered factors. Means in the same column that have the different letter(s) are significantly different using Duncan’s multiple range test at (*p* < 0.05) and ±SE (n = 3) replicates.

**Table 5 molecules-27-02595-t005:** Effect of different treatments of gum arabic (GA) and polyvinylpyrrolidone (PVP) blended with salicylic acid (SA) at different concentrations as a mixture biopolymer coating treatment (GA/PVP–SA) on ethylene and respiration production rate of peach fruit ‘Early Sweet’ during shelf life at ambient air (29 ± 1 °C and air humidity 51%) during two seasons (2020 and 2021).

Treatments		Shelf-Life Period (Days)					
		D _0_	D _2_	D _4_	D _6_	D _8_	D _10_
**Ethylene concentration (C_2_H_4_; mg kg^−1^ h^−1^ FW)**							
	Control	5.71 ± 0.340 ^kl^	24.64 ± 0.583 ^a^	22.25 ± 0571 ^b^	18.52 ± 0.586 ^c^	14.76 ± 0.583 ^de^	11.16 ± 0.554 ^gh^
	GA/PVP–SA 0 mM	5.71 ± 0.340 ^kl^	22.43 ± 0.571 ^b^	19.17 ± 0.591 ^c^	15.14 ± 0.318 ^d^	12.25 ± 0.554 ^fg^	9.23 ± 0.560 ^i^
	GA/PVP–SA 1 mM	5.71 ± 0.340 ^kl^	17.73 ± 0.574 ^c^	13.32 ± 0.886 ^ef^	11.32 ± 0.571 ^gh^	9.38 ± 0.562 ^i^	7.21 ± 0.333 ^jk^
	GA/PVP–SA 2 mM	5.71 ± 0.340 ^kl^	11.17 ± 0.606 ^gh^	9.76 ± 0.307 ^hi^	8.23 ± 0.571 ^ij^	7.28 ± 0.355 ^jk^	5.00 ± 0.330 ^l^
**Respiration evolution (CO_2_; mg kg^−1^ h^−1^ FW)**							
	Control	12.91 ± 0.597 ^fg^	33.16 ± 0.586 ^a^	32.26 ± 0.560 ^a^	28.17 ± 0.640 ^b^	24.03 ± 0.331 ^c^	19.16 ± 0.552 ^e^
	GA/PVP–SA 0 mM	12.91 ± 0.597 ^fg^	27.32 ± 0.568 ^b^	23.82 ± 0.869 ^cd^	22.17 ± 0.346 ^d^	19.23 ± 0.568 ^e^	14.62 ± 0.586 ^f^
	GA/PVP–SA 1 mM	12.91 ± 0.597 ^fg^	22.42 ± 0.577 ^cd^	18.41 ± 0.902 ^e^	14.63 ± 0.853 ^f^	13.36 ± 0.586 ^f^	9.76 ± 0.336 ^hi^
	GA/PVP–SA 2 mM	12.91 ± 0.597 ^fg^	14.54 ± 0.588 ^f^	11.14 ± 0.586 ^gh^	10.52 ± 0.574 ^hi^	8.28 ± 0.353 ^ij^	7.83 ± 0.568 ^j^

The interaction between GA/PVP–SA coating treatments and shelf-life duration in days when both are considered factors. Means in the same column that have different letter(s) are significantly different using Duncan’s multiple range test at *p* < 0.05 and ±SE (n = 3) replicates.

**Table 6 molecules-27-02595-t006:** Modeling of changes in the properties during the shelf life of the treated peach with GA/PVP–SA 2 mM.

Properties	Linear Model (*Y* = *a* ± *bX*) *		
	a (*p* Value)	b (*p* Value)	R^2^
C_2_H_4_	8.99 (0.00)	0.224 (0.23)	0.101
CO_2_	14.11(0.00)	−0.64 (0.00)	0.753
WL%	0.287 (0.231)	1.42 (0.00)	0.988
SBI	0.997 (0.00)	0.0009 (0.015)	0.357
Hue	8.28 (0.00)	1.19 (0.00)	0.814
Firmness	47.86(0.00)	−0.82(0.00)	0.901
TP	88.00(0.00)	−0.82(0.00)	0.755
TF	34.61(0.00)	−0.95(0.00)	0.892
PAL	7.43(0.00)	0.27(0.00)	0.949
PPO	0.26(0.00)	0.01(0.00)	0.933
LOX	0.82(0.00)	0.01(0.00)	0.865
CEL	8.35 (0.00)	0.242(0.002)	0.512
PT	0.27(0.00)	0.02(0.00)	0.931
MDA	0.18(0.00)	0.01(0.00)	0.929
EL%	6.53(0.00)	0.61(0.00)	0.949

* *Y* and *X* denote the dependent (properties) and independent (shelf-life duration) variables, respectively.

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
