# Peer review of "Impacts of Gum Arabic and Polyvinylpyrrolidone (PVP) with Salicylic Acid on Peach Fruit (Prunus persica) Shelf Life"

_molecules, 2022, doi:10.3390/molecules27082595_

Round 1

Reviewer 1 Report

Fruit of peach deteriorates rapidly in edible ripened status, and cannot be stored for long. It is one of the weakest points of the marketing of peach. We can extend the storage period of peaches with different methods of post-harvest technology. By cooling, ensuring a low temperature, and modifying the atmosphere of storeroom, the post-ripening processes of the fruits can be slowed down and better storage can be ensured. These methods are well complemented by edible coatings that allow for even longer shelf life without spoiling the fruit. The article reports on the results of research in this area. The test methods are correct and appropriate conclusions have been drawn from the results. The research results are presented in spectacular figures and usable tables. I support the publication of the article in its current form.

Author Response

Responses to reviewer #1

Fruit of peach deteriorates rapidly in edible ripened status, and cannot be stored for long. It is one of the weakest points of the marketing of peach. We can extend the storage period of peaches with different methods of post-harvest technology. By cooling, ensuring a low temperature, and modifying the atmosphere of storeroom, the post-ripening processes of the fruits can be slowed down and better storage can be ensured. These methods are well complemented by edible coatings that allow for even longer shelf life without spoiling the fruit. The article reports on the results of research in this area. The test methods are correct and appropriate conclusions have been drawn from the results. The research results are presented in spectacular figures and usable tables. I support the publication of the article in its current form.

Dear valued reviewer,

Thank you very much for giving your precious time to review our manuscript and for your very nice comments.

Reviewer 2 Report

The manuscript examines a topical area of research and expands on current knowledge. The English language would benefit from a careful edit to improve the reading quality of the paragraphs. 

The introduction section could be enhanced by using some more recent papers regarding the role of coatings on the shelf life of fruits and vegetables. This would increase the relevance of the introduction to the current situations. So possibly include a few of the references such as:

Sicari, V., Loizzo, M.R., Pellicanò, T.M., Giuffrè, A.M. and Poiana, M. (2020), Evaluation of Aloe arborescens gel as new coating to maintain the organoleptic and functional properties of strawberry (Fragaria × ananassa cv. Cadonga) fruits. Int J Food Sci Technol, 55: 861-870.

Pobiega, K., Igielska, M., WÅ‚odarczyk, P. and Gniewosz, M. (2021), The use of pullulan coatings with propolis extract to extend the shelf life of blueberry (Vaccinium corymbosum) fruit. Int. J. Food Sci. Technol., 56: 1013-1020. 

Melo, N.F.C.B., de Lima, M.A.B., Stamford, T.L.M., Galembeck, A., Flores, M.A., de Campos Takaki, G.M., da Costa Medeiros, J.A., Stamford-Arnaud, T.M. and Montenegro Stamford, T.C. (2020), In vivo and in vitro antifungal effect of fungal chitosan nanocomposite edible coating against strawberry phytopathogenic fungi. Int. J. Food Sci. Technol., 55: 3381-3391.

Zhang, W., Jing, L., Chen, H. and Zhang, S. (2022), NC-1 coating combined with 1-MCP treatment maintains better fruit qualities in honey peach during low-temperature storage. Int. J. Food Sci. Technol., 57: 516-524

I have some concerns over the accuracy of the results in the table. Not because they appear inaccurate but the precision of the equipment would appear to be at its reporting limits in some of the results. What would the researchers feel about the standard deviations observed in Table 3 for instance?

In the description of the results could you be very specific about the reasons why results are observed, sometimes these are vague. For instance 

in line 302 you state "It is thought that high molecular weight poly-
saccharides, such as GA, have huge mechanical properties [42]. " What exactly do you mean by this statement? What are huge mechanical properties and why are they important for your observation? 

In lines 386 - onwards you discuss the role of plant cell walls and shelf life. Could you explain the mechanistic studies which evidence a disruption in cell wall composition and firmness of fruit. What other means could be used to control this?

Author Response

Responses to reviewer #2

-The manuscript examines a topical area of research and expands on current knowledge. The English language would benefit from a careful edit to improve the reading quality of the paragraphs. 

Dear valued reviewer,

Thank you very much for giving your precious time to review our manuscript and for your very nice comments. We would like to mention that the entire manuscript has been checked for potential typos and grammar errors following your suggestion.

-The introduction section could be enhanced by using some more recent papers regarding the role of coatings on the shelf life of fruits and vegetables. This would increase the relevance of the introduction to the current situations. So possibly include a few of the references such as:

  1. Sicari, V., Loizzo, M.R., Pellicanò, T.M., Giuffrè, A.M. and Poiana, M. (2020), Evaluation of Aloe arborescensgel as new coating to maintain the organoleptic and functional properties of strawberry (Fragaria × ananassa Cadonga) fruits. Int J Food Sci Technol, 55: 861-870.
  2. Pobiega, K., Igielska, M., WÅ‚odarczyk, P. and Gniewosz, M. (2021), The use of pullulan coatings with propolis extract to extend the shelf life of blueberry (Vaccinium corymbosum) fruit. Int. J. Food Sci. Technol., 56: 1013-1020. 
  3. Melo, N.F.C.B., de Lima, M.A.B., Stamford, T.L.M., Galembeck, A., Flores, M.A., de Campos Takaki, G.M., da Costa Medeiros, J.A., Stamford-Arnaud, T.M. and Montenegro Stamford, T.C. (2020), In vivoand in vitro antifungal effect of fungal chitosan nanocomposite edible coating against strawberry phytopathogenic fungi. Int. J. Food Sci. Technol., 55: 3381-3391.
  4. Zhang, W., Jing, L., Chen, H. and Zhang, S. (2022), NC-1 coating combined with 1-MCP treatment maintains better fruit qualities in honey peach during low-temperature storage. Int. J. Food Sci. Technol., 57: 516-524

 Thank you very much for providing these very important references. We would like to mention that all the four references were added to the introduction section accordingly.

-I have some concerns over the accuracy of the results in the table. Not because they appear inaccurate, but the precision of the equipment would appear to be at its reporting limits in some of the results. What would the researchers feel about the standard deviations observed in Table 3 for instance?

Thank you very much for your observation regarding the standard division values. The entire tables’ values were checked and confirmed. We agree with you that values of the enzyme activities were decreased to low levels. But, we would like to mention that all the declined values were above the detection limits.

-In the description of the results could you be very specific about the reasons why results are observed, sometimes these are vague. For instance, in line 302 you state "It is thought that high molecular weight poly-saccharides, such as GA, have huge mechanical properties [42]. " What exactly do you mean by this statement? What are huge mechanical properties and why are they important for your observation? 

 The section of results and discussion was checked and corrected accordingly. Also, more explanation to the observations was clarified following your suggestion.

-In lines 386 - onwards you discuss the role of plant cell walls and shelf life. Could you explain the mechanistic studies which evidence a disruption in cell wall composition and firmness of fruit. What other means could be used to control this?

Thank you very much for your valuable comment. The mechanistic evidence was clarified with citing recent references to explain the changes on the cell wall composition and firmness according to your comment.

Reviewer 3 Report

The manuscript entitled “Impacts of Gum Arabic and Polyvinylpyrrolidone (PVP) with Salicylic acid on Peach Fruit (Prunus persica) Shelf Life “ reveals scientific results about extending the shelf life of peaches by coating them with gum arabic an polyvinylpyrrolidone. Nevertheless, in my opinion this manuscript does not fit to special Issue of Molecules entitled Emerging Technologies for Detecting the Chemical Composition of Plant and Animal Tissues and Their Bioactivities

However, I leave this decision to the Editor. As for the scientific aspect, the aim of studies is clearly formulated. The abstract is well written, the keywords are relevant, the introduction and discussion are comprehensive.

However, the minor revision should be taken before manuscript acceptation:

Line 606 - Why water loss %, fruit skin color hue angle, and skin spot browning index were analyzed using one-way instead of two-way analysis?

The authors wrote that each treatment contains 45 fruits (15 fruits per replication), but in the tables they noted that n = 3. My calculations show that Replications x treatments x storage days = 3 x 4 x 6 = 72. Can you clarify this? Additionally, 1 fruit per repetition seems to be not enough. Next time, it would be a good idea to do a minimum of three repetitions of 3 fruits per group per day.

- Line 240 and 251 - "Effect of different treatments of gum arabic (GA) and polyvinylpyrrolidone (PVP) blended with salicylic acid (SA) at different concentrations as a mixture biopolymer coating treatment (GA/PVP–SA) on ethylene and respiration production rate of peach fruit ‘Early Sweet’ during shelf life at ambient air (29 ± 1 °C and air humidity 51%) during two seasons (2020 and 2021)" - Authors should do a two-way analysis and leave either Table 5 or Figure 2.

Author Response

Responses to reviewers #3

The manuscript entitled “Impacts of Gum Arabic and Polyvinylpyrrolidone (PVP) with Salicylic acid on Peach Fruit (Prunus persica) Shelf Life “ reveals scientific results about extending the shelf life of peaches by coating them with gum arabic an polyvinylpyrrolidone. Nevertheless, in my opinion this manuscript does not fit to special Issue of Molecules entitled Emerging Technologies for Detecting the Chemical Composition of Plant and Animal Tissues and Their Bioactivities. However, I leave this decision to the Editor. As for the scientific aspect, the aim of studies is clearly formulated. The abstract is well written, the keywords are relevant, the introduction and discussion are comprehensive.

Dear valued reviewer,

Thank you very much for giving your precious time to review our manuscript and for your very nice comments.

Minor revision should be taken before manuscript acceptation:

Line 606 - Why water loss %, fruit skin color hue angle, and skin spot browning index were analyzed using one-way instead of two-way analysis?

 The section of results and discussion was checked and corrected accordingly. Also, more explanation to the observations was clarified following your suggestion. Regarding statistical analysis, we conducted the non-distractive measurements on the same fruit during all experimental time. Therefore, one-way was adapted for our sample size.

The authors wrote that each treatment contains 45 fruits (15 fruits per replication), but in the tables they noted that n = 3. My calculations show that Replications x treatments x storage days = 3 x 4 x 6 = 72. Can you clarify this? Additionally, 1 fruit per repetition seems to be not enough. Next time, it would be a good idea to do a minimum of three repetitions of 3 fruits per group per day.

Thank you very much for this important observation. The typos was corrected accordingly. In which, for each treatment n=3 (3 expressed on replicates and only each replicate contains 15 fruits).

Line 240 and 251 - "Effect of different treatments of gum arabic (GA) and polyvinylpyrrolidone (PVP) blended with salicylic acid (SA) at different concentrations as a mixture biopolymer coating treatment (GA/PVP–SA) on ethylene and respiration production rate of peach fruit ‘Early Sweet’ during shelf life at ambient air (29 ± 1 °C and air humidity 51%) during two seasons (2020 and 2021)" - Authors should do a two-way analysis and leave either Table 5 or Figure 2.

Thank you very much for your comment. In this regards, we would like to mention that postharvest experiment difference between season (2020 and 2021) does not affects even the repetition of the experiment that the coating treatments have the same effects. Therefore, contribute the season as a factor in the statistical analysis has no significant effect, unless the factor of climate change is included in the fruit behavior during shelf life. We considered your valuable suggestion in our next experiment that are focusing on the seasonal impacts and climatic changes.